# Accelerated Partial Breast Irradiation with Intraoperative Radiotherapy Is Effective in Luminal Breast Cancer Patients Aged 60 Years and Older

**DOI:** 10.3390/jpm12071116

**Published:** 2022-07-08

**Authors:** Michał Falco, Bartłomiej Masojć, Magdalena Rolla, Agnieszka Czekała, Marta Milchert-Leszczyńska, Jolanta Pietruszewska

**Affiliations:** Radiation Oncology Department, West Pomeranian Oncology Center, Strzałowska 22, 71-730 Szczecin, Poland; bmasojc@onkologia.szczecin.pl (B.M.); mrolla@onkologia.szczecin.pl (M.R.); aczekala@onkologia.szczecin.pl (A.C.); mmilchert@onkologia.szczecin.pl (M.M.-L.); jpietruszewska@onkologia.szczecin.pl (J.P.)

**Keywords:** breast cancer, intraoperative radiotherapy, breast recurrence, locoregional relapse

## Abstract

Adjuvant whole breast irradiation (AWBI) improves local control and survival in breast cancer patients after breast-conserving surgery. Between 2010 and 2017, 823 patients ≥ 60 years with ER-positive, Her-2 negative, clinically N0 breast cancer underwent breast-conserving surgery (BCS) at the West Pomeranian Oncology Center. Intraoperative radiotherapy (IORT) with kV photons was applied to 199 (24.2%) patients according to the IORT protocol, and AWBI only was applied to 624 (75.8%). IORT patients in cases with lymph node metastasis, lobular type presence, extensive in situ components, lymphatic vessel invasion, or resection margin < 2 mm, additionally underwent AWBI. Median follow-up was 74 months. There were two (1%) breast relapses in the IORT protocol group and one (0.2%) in the AWBI-only group. In each group, one axillary lymph node relapse was diagnosed (0.5% and 0.2%, respectively). There were two local relapses in the IORT-only group, and they were treated further with BCS and AWBI. Although locoregional relapse-free survival differed between the AWBI-only and IORT protocol groups (98.5% vs. 99.7%, *p* = 0.048), the local control, distant metastasis-free survival, and breast cancer-specific survival were similar. IORT is a reasonable option to avoid AWBI in ER-positive, Her-2 negative, cN0 women with breast cancer aged ≥ 60 years.

## 1. Introduction

Breast-conserving surgery (BCS), which allows breast cancer patients to preserve their breast, is comparable to mastectomy clinical results [1,2]. Obligatory adjuvant whole breast irradiation (AWBI) improved the 10-year local control and 15-year survival rates in patients with or without axillary lymph node metastases [3]. Nevertheless, AWBI requires 3–6 weeks for delivery and may expose patients to potential side effects, such as cardiotoxicity and pulmonary fibrosis.

A subgroup analysis in the Swedish Breast Cancer Group 91 trial (SweBCG 91 RT) showed that AWBI improved the local control for each molecular type [4] and that it was not possible to define patients who could be spared AWBI [5]. Furthermore, randomized trials in early breast cancer patients aged ≥ 65 years were conducted to spare AWBI after BCS. The omission of AWBI significantly increased the local relapse rate, with no difference in distant metastasis free-survival (DMFS) or overall survival (OS) [6,7,8].

However, accelerated partial breast irradiation (APBI) decreases the overall treatment time and minimizes healthy tissue exposure to ionizing energy, which can be delivered as intraoperative electrons [9,10], intraoperative kV photons [11], perioperative brachytherapy [12], high dose rate (HDR) brachytherapy [13,14], or external beam radiotherapy [15,16].

The European Organisation for Research and Treatment of Cancer (EORTC) and the American Society for Radiation Oncology (ASTRO) proposed recommendations to enable the proper qualification for APBI [17,18]. Candidates for APBI can be divided into three groups: “suitable”, “cautionary”, and “unsuitable”, depending on the histological tumor type, diameter, ductal carcinoma in situ (DCIS) presence, excision margin, estrogen receptor (ER) status, lymphatic vessel invasion (LVI), lymph node status, age, and BRCA1 status. In “suitable” patients, compared with AWBI, APBI might lead to a slightly increased local relapse rate [19], with a similar mastectomy rate [13] and comparable OS [10,11,12,13,14,15,16].

The primary aim of the study was to compare the efficacy of the IORT protocol against routine use of AWBI only, in 60-years-and-older breast cancer patients with ER-positive, without Her-2 overexpression tumors, with no clinically involved axillary lymph nodes.

The secondary aim of the study was to compare treatment results between patients who underwent IORT only versus those who received IORT with AWBI.

## 2. Materials and Methods

Between 2010 and 2017, 3876 patients with breast cancer underwent surgery in our hospital. Follow-up data were available for 3800 patients (98%). BCS was performed in 1792 (47.16%) patients with no clinically involved axillary lymph node. Surgery was combined with intraoperative radiotherapy (IORT) and/or AWBI. Eight hundred twenty three (45.9%) women 60 years and older with ER-positive tumors, without Her-2 overexpression, were included in this study. The median follow-up in this group was 72 months (range 5–143 months).

### 2.1. Adjuvant Whole Breast Radiotherapy

Patients with no macroscopic metastases in SLN underwent only whole breast irradiation. Those with macrometastases in SLNs were additionally irradiated to the ipsilateral axillary and supraclavicular lymph nodes. The total dose ranged from 40.05 Gy to 66.00 Gy, with 2.00 Gy to 2.67 Gy per fraction.

### 2.2. IORT Protocol

Since April 2010, IORT administered by a kilovoltage photon emitter (Intrabeam, Carl Zeiss, Oberkochen, Germany) has been used to treat our breast cancer patients. The patients were consulted twice in a multidisciplinary team meeting: before and after surgery. Finally, patients aged ≥ 60 years with no axillary lymph node metastasis; tumor < 3 cm; with no special type, tubular, mucinous type, or molecular luminal tumors were qualified for treatment according to the institutional IORT protocol. The surgical procedure included tumor resection, intraoperative radiological specimen analysis, sentinel lymph node biopsy, and IORT. A dose of 20 Gy using a surface applicator was prescribed. The procedure was conducted as previously described [20]. Following the postoperative histopathological report and a second MDT meeting, patients with no SLN metastases, no extensive ductal in situ component (>20%), no lobular type presence, no LVI, or a resection margin > 2 mm were spared AWBI. The other patients underwent AWBI to the whole breast and/or ipsilateral axillary, supraclavicular lymph nodes. The total dose ranged from 46.00 Gy to 50.00 Gy, with 2.00 Gy per fraction.

### 2.3. Studied Groups

Eight hundred twenty three breast cancer patients 60 years and older with ER-positive tumors with no Her-2 overexpression were analyzed in this study (Figure 1).

Six hundred twenty four (75.8%) patients underwent AWBI only.

One hundred ninety nine (24.2%) patients underwent IORT. Additional AWBI was applied to 97 (48.7%) IORT patients, and 102 (51.3%) received no further irradiation (IORT only). In 18% of patients, there was incompatibility between the institution protocol and the treatment administered [21].

### 2.4. Statistical Analysis

Breast relapse-free survival (IBRFS), locoregional relapse-free survival (LRRFS), DMFS, breast cancer-specific survival (BCSS), and OS were calculated, to compare the 199 patients treated with the IORT protocol against the 624 treated with AWBI only. We used Kaplan–Meier curves for statistical analyses and differences were considered significant when *p* < 0.05.

We analyzed correlations between age (continuous value), primary tumor diameter (pT1a-b vs. pT1c vs. pT2), SLN metastasis (no vs. yes), molecular type (luminal A vs. luminal B), hormonotherapy (no vs. yes), chemotherapy (no vs. yes) use, and IORT protocol vs. AWBI only. The χ^2^ or Fisher’s exact tests or Student’s *t*-test were used to compare differences. The level of significance was set at 5%.

IBRFS, mastectomy free survival (MFS), LRRFS, DMFS, BCSS, and OS in 199 IORT protocol patients were calculated to compare 102 IORT-only patients with those who additionally received AWBI. We used Kaplan–Meier curves and log-rank tests for statistical analyses, and differences were considered significant when *p* < 0.05.

We analyzed correlations between age (continuous value), primary tumor diameter (pT1a-b vs. pT1c vs. pT2), SLN metastasis (no vs. yes), lobular type (no vs. yes), DCIS (no vs. <20% vs. ≥20%), LVI (no vs. yes), resection margin (>2 mm vs. ≤2 mm), molecular type (luminal A vs. luminal B), hormonotherapy (no vs. yes), chemotherapy (no vs. yes) use, and IORT only vs. IORT+AWBI in IORT protocol patients. The χ^2^ or Fisher’s exact tests or Student’s *t*-test were used. The level of significance was set at 5%.

We used multivariate Cox proportional-hazards regression for IBRFS, LRRFS, DMFS, BCSS, and OS in the IORT protocol group, to define independent risk factors for the following variables: primary tumor diameter (pT1a-b vs. pT1c vs. pT2), SLN metastasis (no vs. yes), lobular type (no vs. yes), DCIS (no vs. <20% vs. ≥20%), lymphatic vessel invasion (LVI; no vs. yes), and resection margin (>2 mm vs. ≤2 mm). Differences were considered significant when *p* < 0.05.

Statistical analyses were performed using MedCalc for Windows, version 17.6 (MedCalc Software, Ostend, Belgium).

## 3. Results

Metastasis to the sentinel lymph node (SLN) was not observed in 674 (78.6%) out of 823 analyzed cases. Micrometastases and macrometastases were present in 31 (20.8%) and 118 (79.2%) cases, respectively. Axillary lymph node dissection was performed for 78 (52.3%) patients with SLN metastases and 383 (61.5%) tumors were classified as luminal A and 240 (38.5%) as luminal B. Chemotherapy was administered to 45 (5.5%) patients, and hormonal treatment was administered to 766 (93.1%) cases.

Table 1 shows the treatment results in all 823 analyzed cases. Ipsilateral breast relapses were very rare: two (1%) in the IORT protocol group and one (0.2%) in the AWBI only group. One (0.2%) relapse in the AWBI only group and one (0.5%) case in the IORT protocol group were diagnosed within ipsilateral axillary lymph nodes. IBRFS did not differ significantly between groups, whereas the LRRFS was significantly different in favor of the AWBI-only group (IORT protocol vs. AWBI only 98.5% vs. 99.7%, *p* = 0.048) (Figure 2). Distant metastases were diagnosed in 4 (2%) patients in the IORT protocol group and 22 (3.5%) in the AWBI-only group. Four patients (2%) in the IORT protocol group and 15 (2.4%) in the AWBI-only group died of breast cancer. The DMFS and BCSS did not differ significantly between groups. Fourteen (7%) patients in the IORT protocol group and 78 (12.5%) in the AWBI-only group died during follow-up, which reached statistical significance (93% vs. 87.5%, *p* = 0.047) (Table 1).

The AWBI-only group contained larger tumors (*p* = 0.002), with more frequent SLN metastases (*p* = 0.02), and luminal B histology (*p* = 0.003) when compared to the IORT protocol group. There was no statistical difference in age. Patients in the AWBI group more frequently underwent adjuvant chemotherapy (*p* = 0.035), and omission of hormonal treatments was also observed in this group (*p* = 0.013) when compared to the IORT protocol (Table 2).

There were two (2%) relapses in the ipsilateral breasts of those with IORT only compared with no relapse in the IORT+AWBI patients. The relapses were truly local: one was invasive (after 56 months) and one was preinvasive (after 9 months). Both relapsing patients underwent BCS and AWBI, and therefore no patients in the APBI protocol group underwent salvage mastectomy. One patient with a relapse (1%) was diagnosed with ipsilateral supraclavicular lymph nodes in the IORT+AWBI group and none were noted in the IORT only group. IBRFS and LRRFS were not significantly different between groups. Four patients (3%) in the IORT+AWBI group, and none in the IORT-only group, experienced distant metastases and died of breast cancer. During follow-up, seven (6.9%) patients in the IORT-only group and three (3%) in the IORT+AWBI group died of other reasons. Three patients died of secondary non-breast cancers. DMFS (IORT only vs. CBRT 100% vs. 95.9%, *p* = 0.04) and BCSS (IORT only vs. CBRT 100% vs. 95.9%, *p* = 0.04) were significantly different between groups in favor of IORT only. Nevertheless, the OS did not differ significantly between groups (Table 3).

All patients with LVI and positive SLNB, and all but one patient with a lobular or an extensive DCIS (>20%) component, underwent adjuvant whole breast irradiation in the IORT protocol group. The larger the tumor, the more frequently adjuvant irradiation was administered. Most patients with a resection margin < 2 mm underwent AWBI. There were no differences in age or luminal type tumors between groups. Patients in the IORT + AWBI group more frequently underwent adjuvant chemotherapy when compared to the IORT-only group (*p* = 0.02). Most patients in both groups received hormonotherapy (Table 4).

Multivariate analysis did not show any independent risk factor for IBRFS, LRRFS, DMFS, BCSS, and OS in the APBI protocol group.

## 4. Discussion

We found very good IBRFS in the analyzed group (99.6%) and IORT protocol (99.0%), with a median follow-up of 72 months. The PRIME II trial recruited women aged ≥ 65 years with T1-2N0 and ER-positive breast cancer to verify the role of AWBI. The 5-year IBRFS was 98.7%, compared with 95.9% for those without AWBI, and the 5-year lymph node recurrence-free survival rates were 99% and 96%, respectively [6]. The CALGB 9343 trial analyzed women aged ≥ 70 years with T1N0 and ER-positive breast cancer. All patients underwent hormone therapy and the 10-year LRRFS was 98% for those administered AWBI vs. 90% in those spared AWBI [7]. Martelli et al. presented non-randomized data on ER-positive breast cancer patients with tumors < 3 cm and aged ≥ 70 years: the 15-year breast relapse rate was 3.9% after AWBI and 9.2% with no AWBI [8]. In a population-based study by Kurian et al., the addition of AWBI to hormone therapy in T1-2N0, ER-positive women aged ≥ 70 years improved the LRRFS from 94% to 99% [22]. In another retrospective analysis of breast cancer patients aged ≥70 years, the LRRFS at 10 years was 97% [23]. Furthermore, the IBRFS and LRRFS were improved by adding AWBI to hormonal treatment in younger patients [24,25]. Adjuvant hormonal therapy with no AWBI resulted in IBRFS and LRRFS of 94–96% with 5–6 years follow-up [6,22], and 90% after 10–15 years [7,8]. AWBI added to hormonal treatment increased those values to 98–99% with 5–10 years of follow-up [6,7,22], and 96% after 15 years [8]. The results in both the AWBI-only and IORT protocol group are comparable to those in studies with 5–6 years follow-up. Nevertheless, the longer the follow-up, the more frequent locoregional relapses in patients with no AWBI were observed [7,8]. In the IORT-only group, the area of the tumor bed was irradiated, with a negligible dose to the remaining breast tissue. The question of locoregional relapses with follow-ups longer than 5–6 years is still open, as trials on APBI report results with a 5-year follow-up [10,11,13,15,16].

The TARGIT trial compared BCS combined with AWBI or IORT [11]. After BCS and IORT, patients qualified for AWBI following a pathological report. The indication for AWBI was metastasis in SLN, an unexpected extensive in situ component or lobular histology, and extensive LVI [11]. In our IORT protocol group, 97 (48.7%) patients qualified for AWBI, in contrast to 21.6% in the TARGIT trial [11]. We used the same device and application method as that in the TARGIT trial. In the IORT protocol group, the IBRFS and LRRFS were 99% and 98.5% in our study compared with 97.9% and 97.9% in the TARGIT trial [11]. In the ELIOT trial, where intraoperative electron irradiation was used with no AWBI, an IBRFS of 98.1% and LRRFS of 97.8% were presented in the “suitable” group of patients, according to EORTC recommendations [10]. The HDR brachytherapy used for APBI in the “suitable” group, according to EORTC, resulted in a low relapse rate, a 5-year IBRFS of 98.6%, and a 5-year lymph node relapse-free survival rate of 99.5% [13]. The IMPORT LOW trial and a trial conducted at the University of Florence utilized external beam irradiation as APBI for breast cancer patients. In the IMPORT LOW trial, “tangential” fields covering almost one second of the breast were used, with a 5-year IBRFS of 99.5% and LRRFS of 99.2% [15]. The University of Florence trial used APBI with intensity modulated radiotherapy [16]. The 5-year IBRFS was 98.5% and the LRRFS was 98.5%, and for patients aged ≥ 70 years, the rates were 98.1% for both [16,26]. Almost 90% of patients in those trials received hormonal treatment. This might be an important modality of adjuvant treatment to prevent locoregional relapses outside the tumor bed.

In our study, there was a significant difference in the LRRFS between the AWBI-only and IORT protocol groups, although the number of events was very low and comparable with other studies. This might be explained by the fact that almost 97% and 92% of patients in the IORT protocol and AWBI groups, respectively, underwent hormone therapy.

There were two true local relapses in the IORT-only group, with neither in breast outside tumor bed nor ipsilateral lymph node relapses. No mastectomy was performed in the IORT-only group. The advantage of intraoperative kV photon usage is the possibility to administer AWBI either as an adjuvant treatment or as part of salvage therapy. In the CALGB 9343 trial, the 10-year MFS was 90% in the tamoxifen only group and 98% if AWBI was applied [7]. A 5-year MFS of 99.8% was reported in the HDR brachytherapy trial, which was comparable with our data [13].

Differences in the LRRFS between the AWBI-only and IORT protocol groups had no impact on the DMFS and BCSS. This might be explained by the fact that isolated cases with no lymph node relapses and breast relapses do not lead to distant metastases, as we reported previously [27].

There were no significant differences in the DMFS between the IORT protocol and AWBI groups in our study. AWBI also had no impact on the DMFS in AWBI vs. no AWBI studies [6,7] and APBI studies [10,13,14,15,16,26]. There were no distant metastases or breast cancer deaths in the IORT-only group, compared with four deaths in the IORT+AWBI group. This might be related to the qualification protocol. The IORT + AWBI group consisted of patients with worse prognostic factors, i.e., SLN metastases, pT2, or LVI positivity (Table 3). The BCSS in our study was very high in the analyzed group and there was no difference between the IORT protocol and AWBI-only groups. APBI trials presented comparable results, demonstrating no differences when compared with AWBI [11,14,16]. In the CALGB9343 and PRIME II trials, and studies by Martelli et al. and Kurian et al., the AWBI did not improve the BCSS in women aged ≥ 65 [5,6,7,22]. In contrast, the AWBI after BCS improved the BCSS in the Early Breast Cancer Trialist Collaborative Group metanalysis [3].

The OS was worse in the AWBI-only group compared with the IORT protocol group in our study. The PRIME II and CALGB 9343 randomized trials reported no difference in the OS in ER-positive, N0 women aged ≥ 65 years [6,7]. However, data on AWBI use from retrospective analyses of ER-positive, N0 women aged ≥ 65 years are conflicting. Some studies showed no differences [8,28] and others showed differences in favor of AWBI use [22,29,30,31]. The greatest benefit of AWBI use was observed when no hormonal treatment was administered [22,30,31]. Furthermore, APBI resulted in a comparable OS when compared with AWBI [10,13,14,15,16,26]. It seems, in that group of patients, factors other than cancer mortality started to play a role with longer follow-ups.

The IORT procedure is performed during breast operations. Contrary to other irradiation methods, it does not require further multiple visits to hospital. For example, AWBI takes 3–5 weeks to be delivered, with brachytherapy for 5 consecutive days and linac-based APBI for at least 9–10 days. Furthermore, IORT does not expose patients to the potential side effects of AWBI, i.e., cardiotoxicity or pneumonitis.

The decreased OS in the AWBI-only group when compared to the IORT protocol might be correlated to the worse prognostic factors observed in the group characteristics. Those patients also more frequently underwent adjuvant chemotherapy and less frequently underwent hormotherapy. The decreased BCSS in the IORT + AWBI group when compared to IORT-only was caused by the qualification criteria of omitting further AWBI.

The results presented in this study might be biased by a few factors: the retrospective design, patient groups treated in one center, and the relatively low number of IORT patients when compared to multicenter prospective trials, i.e., PRIME II—1326 patients, CALGB 9343—636 patients, TARGIT—1721 patients [6,7,11]. On the other hand, in this study, we present “real life” results of breast cancer radiotherapy with moderately long follow-up times.

## 5. Conclusions

The implementation of the IORT protocol for women aged ≥ 60 years does not decrease most survival endpoints, although it might negatively influence the locoregional relapse rate. The protocol we used allowed 102 (12.4%) breast cancer patients to be spared AWBI, with no higher risk of clinically meaningful relapses, no exposition to potential side effects (i.e., cardiotoxicity or pneumonitis), and comparable overall survival.

Therefore, IORT is a reasonable option to avoid AWBI in ER-positive, Her-2 negative women with breast cancer aged ≥ 60 years.

## Figures and Tables

**Figure 1 jpm-12-01116-f001:**
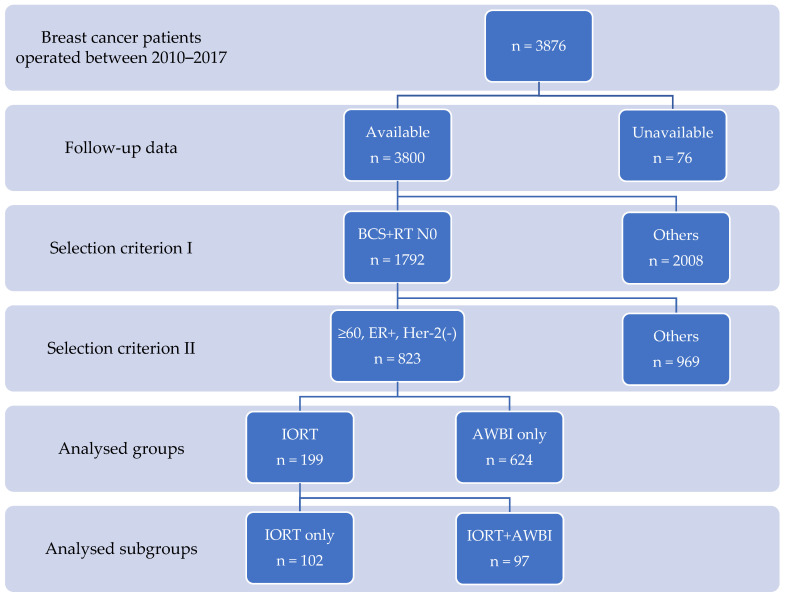
Patient selection criteria. BCS + RT N0: breast-conserving surgery with radiotherapy in clinically node negative patients; ≥60 y, ER+, Her-2(-): ≥60 years old patients who were estrogen positive, no Her-2 overexpression breast cancer patients; IORT: patients who underwent intraoperative radiotherapy; AWBI only: patients who underwent adjuvant whole breast radiotherapy only; IORT only: patients who underwent intraoperative radiotherapy only; IORT + AWBI: patients who underwent intraoperative radiotherapy and adjuvant whole breast radiotherapy.

**Figure 2 jpm-12-01116-f002:**
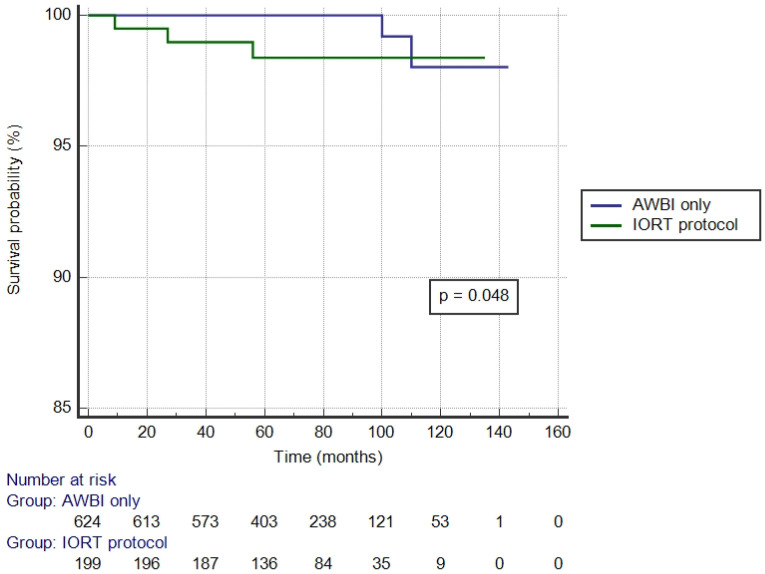
Locoregional relapse-free survival in the analyzed group.

**Table 1 jpm-12-01116-t001:** Treatment results in the analyzed group.

	*n*	IBRFS	*p*	LRRFS	*p*	DMFS	*p*	BCSS	*p*	OS	*p*
ALL	823	99.6%		99.4%		96.8%		97.7%		88.8%	
IORT protocol	199	99.0%	NS	98.5%	0.048	98.0%	NS	98.0%	NS	93.0%	*p* = 0.047
AWBI only	624	99.8%		99.7%		96.5%		97.6%		87.5%	

IBRFS—in breast relapse-free survival; LRRFS—locoregional relapse-free survival; DMFS—distant metastasis-free survival; BCSS—breast cancer-specific survival; OS—overall survival; IORT protocol—patients who underwent intraoperative radiotherapy; AWBI only—patients who underwent only adjuvant whole breast radiotherapy; NS—non-significant.

**Table 2 jpm-12-01116-t002:** Characteristics of patients analyzed in study.

		IORT Protocol	AWBI only	*p*
		199	624	
Age (years)	Mean value	67.4 (66.6–68.2)	67.6 (67.1–68.1)	NS
pT	1a-b	85 (10.4%)	179 (21.9%)	0.002
	1c	94 (11.5%)	314 (38.5%)	
	2	20 (2.5%)	124 (15.2%)	
pSLN	Negative	174 (21.1%)	500 (60.7%)	0.02
	Positive	25 (3%)	124 (15.1%)	
Molecular type	Luminal A	115 (18.5%)	268 (43%)	0.03
	Luminal B	46 (7.4%)	194 (31.1%)	
Hormonotherapy	No	6 (0.7%)	51 (6.2%)	0.013
	Yes	193 (23.5%)	573 (69.6%)	
Chemotherapy	No	5 (6.1%)	40 (4.9%)	0.035
	Yes	194 (23.6%)	584 (70.1%)	

IORT protocol—breast cancer patients who underwent intraoperative radiotherapy; AWBI only—breast cancer patients who underwent only adjuvant whole beast radiotherapy; pSLN—pathological sentinel lymph node; NS—non-significant.

**Table 3 jpm-12-01116-t003:** Treatment results in patients treated with intraoperative radiotherapy (subgroup analysis).

	*n*	IBRFS	*p*	LRRFS	*p*	DMFS	*p*	BCSS	*p*	OS	*p*
IORT only	102	98.0%	NS	98.0%	NS	100.0%	0.04	100.0%	0.04	93.1%	NS
IORT + AWBI	97	100.0%		99.0%		95.9%		95.9%		92.8%	

IBRFS—in breast relapse-free survival; LRRFS—locoregional relapse-free survival; DMFS—distant metastasis-free survival; BCSS—breast cancer-specific survival; OS—overall survival; IORT only—intraoperative radiotherapy with no adjuvant whole breast irradiation; IORT+AWBI—intraoperative radiotherapy combined with adjuvant whole breast irradiation. NS—non-significant.

**Table 4 jpm-12-01116-t004:** Characteristics of patients treated with the IORT protocol.

		IORT Only	IORT + AWBI	*p*
		102	97	
Age (years)	Mean value	68.18 (66.89–69.48)	66.55 (65.54–67.55)	NS
pT	1a–b	52 (26.1%)	33 (16.6%)	0.005
	1c	44 (22.1%)	50 (25.1%)	
	2	6 (3%)	14 (7%)	
pSLN	Negative	102 (51.3%)	72 (36.2%)	<0.0001
	Positive	0 (0%)	25 (12.5%)	
Lobular	Negative	101 (50.7%)	81 (40.7%)	0.0001
	Present	1 (0.5%)	16 (8%)	
Molecular type	Luminal A	56 (34.8%)	59 (36.6%)	NS
	Luminal B	21 (13%)	25 (15.5%)	
LVI	No	102 (51.3%)	86 (43.2%)	0.0005
	Yes	0 (0%)	11 (5.5%)	
DCIS	No	78 (39.2%)	44 (22.1%)	<0.0001
	<20%	23 (11.6%)	38 (91.1%)	
	>20%	1 (0.5%)	15 (7.5%)	
Margin	>2 mm	88 (44.2%)	57 (28.6%)	<0.0001
	≤2 mm	14 (7%)	40 (20.1%)	
Hormonotherapy	No	3 (1.5%)	3 (1.5%)	NS
	Yes	99 (49.7%)	94 (47.2%)	
Chemotherapy	No	102 (51.3%)	92 (46.2%)	0.02
	Yes	0	5 (2.5%)	

IORT only—intraoperative radiotherapy with no adjuvant whole breast irradiation; IORT + AWBI—intraoperative radiotherapy combined with adjuvant whole breast irradiation; pSLN—pathological sentinel lymph node; LVI—lymphatic vessels invasion; DCIS—ductal carcinoma in situ; NS—non-significant.

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
