# Peer review of "Accelerated Partial Breast Irradiation with Intraoperative Radiotherapy Is Effective in Luminal Breast Cancer Patients Aged 60 Years and Older"

_jpm, 2022, doi:10.3390/jpm12071116_

Round 1

Reviewer 1 Report

The manuscript discusses about a special radiotherapy for breast cancer patients. The manuscript contains valuable information; however, it is not an easy read due to incoherent use of acronyms (a list of acronyms will help) and overrepresentation of data. It also misses a hypothesis/specific aims statement. Furthermore, the methods section needs a lot of improvement to understand the study better. Some suggestions may improve the readability.

Abstract:

Line 14: Please provide the details of the center.

Line 14: Please make the treatment groups mutually exclusive as much as possible to clear the confusion. For example, if 199 patients out of 823 patients got the IORT, please mention the alternative treatment the remaining patients got.

Line 15: Please add the full meaning of APBI and define APBI IORT.

Materials and methods:

Ø  This section will be better if segmentized.

Ø  Some parts should go into results section.

Ø  Please include a section of APBI protocol.

Ø  Please include a section statistical analysis.

Results:

Ø  A demographics of the patients should be added.

Ø  A treatment scheme will help to understand all the treatment groups.

Ø  Please include statistical details in the legends for all figures and tables.

Ø  Starting point of Y-axis on figures 1 and 2 should be higher. Please include about how the survival probability was calculated in the Methods section. What is significance of including the figures since the data are already on the tables?

Ø  Please include the meaning of NS and mentioned the groups compared for all p values.

Ø  The reviewer did not understand the organization of the results section and felt that it will be improved a lot if the whole section is reorganized based on the hypothesis or key findings.

Discussion:

Discussion can be precise. Please include the key findings and literature relevant to the key findings. Also, specifically discuss why IORT will be better than others.

Conclusion:

Similar to discussion section, the conclusion part should specifically mention why IORT is better than AWBI in terms of survival, relapse, and toxicity.

Author Response

Thank  you for reviewing the manuscript. We have made major revisions according to reviewers suggestions which include:

  1. reducing the overrepresentation of data in materials, methods and results,
  2. clarification of study aims,
  3. addition of Figure 1 which we hope helps to understand study design,
  4. deletion of some parts of discussion that related to removed data ,
  5. modification of patient group names i.e: CBRT to IORT+AWBI or APBI to IORT,
  6. change in Figure 1 – new name Figure2 Locoregional relapse-free survival in analysed group,
  7. removal of Table1 according to change in results part,
  8. division of Table 2 into two table with new names Table 1 and Table3,
  9. addition of table: characteristics of patients analysed in study – table 2.
  10. removal of former Figure2, as it presented unnecessary data,
  11. deletion of 2 references (nr 22, 23) following removal of part of discussion and change in numbers in references .

Response to comments:

We have added the acronym list. We have reduced amount of presented data. We clarified study aims. We have modified methods section to improve the readability

Abstract:

We have provided details of the centre.

We changed tha abstract according to reviewers suggestions.

Materials and methods:

We segmentized the chapter according to suggestions and followed further recommendations.

Results:

We changed following reviewer suggestions.

Discussion:

We changed following reviewer suggestions.

We have added a paragraph (penultimate) according to reviewers suggestion

Conclusion:

We changed following reviewer suggestions.

Reviewer 2 Report

#Table 2 needs to be more readable. The categorization of the table is confusing, and there is an error in spacing(APB Iprotocol)

#line 81> resection margin <2 mm were spared AWBI. 
Shouldn't the resection margin be more than 2mm to avoid AWBI? The inequality notation is weird.

Author Response

Thank  you for reviewing the manuscript. We have made major revisions according to reviewers suggestions which include:

  1. reducing the overrepresentation of data in materials, methods and results,
  2. clarification of study aims,
  3. addition of Figure 1 which we hope helps to understand study design,
  4. deletion of some parts of discussion that related to removed data ,
  5. modification of patient group names i.e: CBRT to IORT+AWBI or APBI to IORT,
  6. change in Figure 1 – new name Figure2 Locoregional relapse-free survival in analysed group,
  7. removal of Table1 according to change in results part,
  8. division of Table 2 into two table with new names Table 1 and Table3,
  9. addition of table: characteristics of patients analysed in study – table 2.
  10. removal of former Figure2, as it presented unnecessary data,
  11. deletion of 2 references (nr 22, 23) following removal of part of discussion and change in numbers in references .

Response to comments:We divided the table following suggestion.

We corrected the line 81.

Reviewer 3 Report

First, thank you for allowing me to review the paper "Accelerated partial breast irradiation with intraoperative radiotherapy is effective in luminal breast cancer patients aged 60 years and older" and congratulations to the authors for their work.

However, I would like to make a few remarks:

From the point of view of errata, I noticed that in line 74, in "Tumor >3cm" it should be <3cm to be consistent with Table 1. In addition, in line 81, it is similar with "<2mm", which should be >2mm to comply with the protocol of avoiding whole-breast radiotherapy

The format of Table 2 is confusing as it is not very intuitive to differentiate the two groups of patients: APBI protocol+ Non-APBI and those included in the APBI protocol.

Table 4 does not exist (line186) and from the context it seems to refer to Table 3.

Discussion:

In the first paragraph of the discussion, mention is made of local and locorregional relapses. The authors assert that the relapse rate in their cohort of patients is much lower than that presented in the literature (refs 4 and 22). But they omit that in these studies the follow-up was 15 and 8 years respectively, higher than the median follow-up of this study (6 years). Therefore, I consider that the comparison without taking into account this issue should be revised or, at least mentioned.

In the discussion, I miss a critical reading of the results, of the possible biases, what it contributes to scientific knowledge, what possible factors can make the findings different or similar to those of other studies...more than simply making a crude comparison of the results of this and other works.

Author Response

Thank  you for reviewing the manuscript. We have made major revisions according to reviewers suggestions which include:

  1. reducing the overrepresentation of data in materials, methods and results,
  2. clarification of study aims,
  3. addition of Figure 1 which we hope helps to understand study design,
  4. deletion of some parts of discussion that related to removed data ,
  5. modification of patient group names i.e: CBRT to IORT+AWBI or APBI to IORT,
  6. change in Figure 1 – new name Figure2 Locoregional relapse-free survival in analysed group,
  7. removal of Table1 according to change in results part,
  8. division of Table 2 into two table with new names Table 1 and Table3,
  9. addition of table: characteristics of patients analysed in study – table 2.
  10. removal of former Figure2, as it presented unnecessary data,
  11. deletion of 2 references (nr 22, 23) following removal of part of discussion and change in numbers in references .

Response to comments:We changed >3 to <3.

We changed the line 81.

We divided the table following suggestion.

We changed tables names and corrected the mistake.

According to reviewers suggestions regarding overrepresentation of data we removed part of materials, results and in consequence part of discussion in which we analysed correlation with references 4 and 22.

We have added a paragraph (the last one) according to reviewers suggestion.

Reviewer 4 Report

Comments regarding the manuscript titled: “Accelerated partial breast irradiation with intraoperative radio-2 therapy is effective in luminal breast cancer patients aged 60 3 years and older”

The authors evaluated the efficacy of the accelerated partial breast irradiation (APBI) protocol for in group of breast cancer patients with specific characteristics (n=823). The authors present intraoperative radiotherapy (IORT) as an acceptable option to adjuvant whole breast irradiation (AWBI) in women with ≥60 years of age with estrogen receptor positive, Her-2 negative, cN0 breast cancer.

Materials and Methods section: The authors should consider the use of subheadings to improve the readability of the section. The first paragraph reads as a Results section instead of describing the methods used. Maybe include a flowchart to visually explain the patient selection and further stratification into study groups.

Line 82: Further explain the incompatibility exposed on this statement.

Results section: Overall this section is difficult to follow, the authors can improve this by included subheadings for each main finding or each main analysis.

Line 131, 134, and 136: There are statements regarding differences in tumor diameter; however, this variable is not included on Table 1.

Line 131-132: The statistical significance for molecular subtype in IBRFS is >0.05 (p=0.051). Please check this statement.

Tables: The superscripts are not defined on the footnote of the tables. If the authors are using them for the abbreviations, they should include them on the footnote or just include the abbreviations without the superscripts.

Table 2: Classification of the participants by treatment is difficult to visualize with the current format of the table. Please check the table formatting for readability.

Figure 1: Include the number of participants per group on the legend or the figure caption.

Figure 2: The figure caption is incomplete. Also, include the number of participants per group on the legend or the figure caption.

Lines 162-164: These data are not included on any of the tables. Verify this statement.

Line 186: The authors cite Table 4 but there are only three tables included on the manuscript.

Author Response

Thank  you for reviewing the manuscript. We have made major revisions according to reviewers suggestions which include:

  1. reducing the overrepresentation of data in materials, methods and results,
  2. clarification of study aims,
  3. addition of Figure 1 which we hope helps to understand study design,
  4. deletion of some parts of discussion that related to removed data ,
  5. modification of patient group names i.e: CBRT to IORT+AWBI or APBI to IORT,
  6. change in Figure 1 – new name Figure2 Locoregional relapse-free survival in analysed group,
  7. removal of Table1 according to change in results part,
  8. division of Table 2 into two table with new names Table 1 and Table3,
  9. addition of table: characteristics of patients analysed in study – table 2.
  10. removal of former Figure2, as it presented unnecessary data,
  11. deletion of 2 references (nr 22, 23) following removal of part of discussion and change in numbers in references .

Response to comments:Materials and Methods section:

We added the subheadings and flowchart (Figure1).

We changed line 82 according to reviewer’s suggestion.

Results:

According to other reviewers suggestions we reduced the amount of presented data. In consequence lines 131-136 and Table 1 were removed.

We removed superscripts in tables.

Figure 1: Included the number of participants per group on the legend or the figure caption, we changed the number

Table 2 : we divided in two tables.

Figure 2: was removed, as it presented unnecessary data

Lines 162-164: We have added additional demographic table – Table 2.

Line 186: we corrected.

Round 2

Reviewer 1 Report

Thank you for the edits.

Author Response

Thank you for your acceptance. Following other reviewers suggestions we performed minor revision:

  1. We have created clear copy in pdf format according, word format is a tracked copy (Reviewer 3)
  2. Bullets were used for better visualisation of sections in materials and methods (Reviewer 4)
  3. Figure 1 was edited (Reviewer 4)
  4. We moved part of data from materials and methods section to results section (Reviewer 3)
  5. We have edited discussion section (Reviewer 3)

Reviewer 3 Report

It would have been essential to have presented the PDF of the article without the modifications marks of the Word format, since it is extremely difficult to read the text fluently and this makes it impossible to review it properly.

A substantial part of the data presented in material and methods would correspond to results.

The discussion continues to be practically an enumeration of the results of different studies without any real connecting thread between them that would give it structure, homogeneity and precision.

Author Response

Thank you for your comments. Following your and other reviewers suggestions we performed minor revision:

  1. We have created clear copy in pdf format according, word format is a tracked copy (Reviewer 3)
  2. Bullets were used for better visualisation of sections in materials and methods (Reviewer 4)
  3. Figure 1 was edited (Reviewer 4)
  4. We moved part of data from materials and methods section to results section (Reviewer 3)
  5. We have edited discussion section (Reviewer 3)

Reviewer 4 Report

I would like to thank the authors for accepting and including my comments and suggestions on their manuscript. I believe that it has improved in terms of clarity and readability. However, there are still minor details that I would like to point out:

-Line 99: When the authors explain the study groups, maybe the use of bullets would be helpful for better visualization of the section.

-Figure 1: The caption should be located below the figure. Also, capital letters are more appropriate for selection criteria in the flowchart. For example, Breast cancer patients operated between 2010-2017. Also, on the boxes begin the words with capital letter and make sure that the font is consistent with the rest of the document. Instead of using pts for patients, the authors could use n=3876 since it is known that the study participants are breast cancer patients.

-Figure 2 seems to be duplicated, please verify.

Author Response

Thank you for your comments. Following your and other reviewers suggestions we performed minor revision:

  1. We have created clear copy in pdf format according, word format is a tracked copy (Reviewer 3)
  2. Bullets were used for better visualisation of sections in materials and methods (Reviewer 4)
  3. Figure 1 was edited (Reviewer 4)
  4. We moved part of data from materials and methods section to results section (Reviewer 3)
  5. We have edited discussion section (Reviewer 3)
  6. The caption is located above the figure according to journal requirements

    Figure 2 was veryfied (follow the clear pdf copy)